**Two-dimensional monitoring of air pollution in Madrid using a MAXDOAS-2D**
**instrument**
David Garcia-Nieto[1, 2], Nuria Benavent[1, 2], Rafael Borge[2] and Alfonso Saiz-Lopez[1]
[1] Department of Atmospheric Chemistry and Climate, Institute of Physical Chemistry
Rocasolano, CSIC, Madrid 28006, Spain
[2] Universidad Politécnica de Madrid, UPM, 28006 Madrid, Spain
*Corresponding author: Alfonso Saiz-Lopez (a.saiz@csic.es)
**Abstract**

14       Trace gases play a key role in the chemistry of urban atmospheres. Therefore,

knowledge about their spatial distribution is needed to fully characterize air quality in
urban areas. Using a new Multi-AXis Differential Optical Absorption Spectroscopy
(MAXDOAS)-2D instrument, along with an inversion algorithm (bePRO), we report the
first two-dimensional maps of nitrogen dioxide ($NO_2$) and nitrous acid (HONO)
concentrations in the city of Madrid, Spain. Measurements were made during two
months (May 6 –July 5 2019) and peak mixing ratios of 12 ppbv and 0.7 ppbv for $NO_2$
and HONO, respectively, were observed in the early morning in the southern part of
the downtown area. We found good general agreement between the MAXDOAS-2D
mesoscale observations -which provide a typical spatial range of a few kilometers- and
the in-situ measurements provided by Madrid´s air quality monitoring stations. In
addition to vertical profiles, we studied the horizontal gradients of $NO_2$ in the surface

layer by applying the different horizontal light path lengths in the two spectral regions included in the $NO_2$ spectral analysis: ultraviolet (UV, at 360 nm) and visible (VIS, 477 nm). We also investigate the sensitivity of the instrument to infer vertically-distributed information on aerosol extinction coefficients and discuss possible future ways to improve the retrievals. The retrieval of two-dimensional distributions of trace gas concentrations reported here provides valuable spatial information for the study of air quality in the city of Madrid.

**1 Introduction**

Air pollution in urban areas has become a concern in our society because it represents a major risk to human health and the environment (WHO, 2019). Air quality is often expressed as the state of air pollution in terms of gaseous pollutant concentrations as well as size and number of particulate matter that may affect human health, ecosystems and climate (Monks et al., 2009). Integral understanding of air pollution requires knowledge about the sources, pollutants, chemical composition and spatial distribution, and their transport phenomena in the atmosphere (EEA, 2019).

Madrid, Spain, has suffered from severe air pollution in recent years, with episodes of large nitrogen dioxide ($NO_2$) and ozone ($O_3$) concentrations. In an effort to control and reduce high pollution events, the local government has enforced some traffic restriction measures (Izquierdo et al., 2020) and has set up several in-situ air quality monitoring stations over the city's metropolitan area. These in-situ instruments -as of today- cannot measure some important trace gases present in the atmosphere and their values are only representative of the immediate surrounding of the instruments and at surface level. There is therefore a need for mesoscale analysis (both in horizontal and vertical, in the order of 10 km) of urban air pollution that could

complement the in-situ measurements. With this aim, we have deployed a Multi AXis Differential Optical Absorption Spectroscopy (MAXDOAS) instrument for air pollution measurements in Madrid. MAXDOAS is a widely used technique for the detection of trace gases in the atmosphere and it is based on the wavelength dependent absorption of scattered sunlight by atmospheric constituents (Platt and Stutz, 2008). In addition to routinely monitored species such as $NO_2$ and $O_3$, MAXDOAS provides mesoscale measurements of other trace gases that are relevant to understand atmospheric chemistry, such as nitrous acid (HONO), formaldehyde (HCHO) or glyoxal (CHOCHO). Over the past few years, we have reported trace gas measurements in Madrid using the MAXDOAS technique (Wang et al., 2016; Garcia-Nieto et al., 2018; Benavent et al., 2019) as well as pollutants trend analysis and chemical transport modelling (Borge et al., 2018; Cuevas et al., 2014; Saiz-Lopez et al., 2017).

For this work, a new two-dimensional MAXDOAS instrument (which will be described in Sect. 3 and will be hereafter referred to as MAXDOAS-2D) has been built, tested and set up to take continuous measurements in Madrid. This instrument represents a follow-up development to our previous one-dimensional instrument (MAXDOAS-1D, see Wang et al., 2016) that incorporates the capability of moving in the azimuthal dimension, therefore allowing the collection of spectra pointing at any angular direction. This additional capability allows the measurement of both the horizontal and vertical trace gas (e.g. $NO_2$) distribution throughout the city and in turn the generation of two-dimensional maps of trace gas concentrations. Several works using two-dimensional MAXDOAS instruments have been carried out in recent years (e.g. Ortega et al., 2015, Yang et al., 2019, Schreier et al., 2020, Dimitropolou et al., 2020). These studies were mostly focused on mapping the $NO_2$ distribution in urban environments and assessing its role for air quality monitoring.

Here we present two months of MAXDOAS-2D measurements of scattered
sunlight spectra. The measurements were taken from May 6, 2019 to July 5, 2019, with
focus on the evaluation of NO$_2$ vertical concentration profiles and the characterization
of horizontal light path lengths. We also provide the retrieval of HONO as an example
of the potential of the MAXDOAS-2D to provide spatial information also on other trace
gases relevant to urban atmospheric chemistry. An assessment of the relation
between the MAXDOAS data and the in-situ measurements of NO$_2$ in the city was
carried out. Sect. 2 provides details of the DOAS technique while Sect. 3 describes the
experimental setup. The inversion methods and the atmospheric parameters chosen
for the analysis are detailed in Sect. 4. The two-dimensional NO$_2$ and HONO
distributions, an evaluation of the light path geometries, along with their relative
probabilities, and an assessment of horizontal mixing ratio gradients near the surface
are discussed in Sect. 5.

**2 Brief introduction to the DOAS method**

The DOAS basic idea is described by the Beer-Lambert law, which models the
exponential attenuation of spectral irradiance when it traverses a certain sample that
contains some absorbers:

$$I(\lambda, L) = I_0(\lambda)\, exp\left(-\sum_i \int_0^L \sigma_i(\lambda)\, \rho_i(s)ds\right) \tag{1}$$

where $\lambda$ is wavelength, $\sigma_i$ and $\rho_i$ stand for -respectively- the absorption cross section
and concentration of a given absorber $i$ along the path, while the pair $I_0$ and $I$
represent the spectral irradiances at the beginning and end of the process at study.
The absorption processes are integrated over the photon paths (with infinitesimal
path $ds$) and summed over every present absorber (Platt and Stutz, 2008).

Specifically, the MAXDOAS technique is based on the study of the differential

spectral absorption structures that are produced in the measured scattered sunlight
spectra (Hönninger et al., 2004; Plane and Saiz-Lopez, 2006; Platt and Stutz, 2008). The
main principle is based on identifying the narrowband absorption features within the
measured optical density taking out the broadband optical density, mainly generated
by Rayleigh and Mie scattering, as well as by instrumental effects. On the other hand,
an analogous process is done on the trace gases absorption cross sections by means
of filtering out the broadband spectral features, hence producing the so-called
differential absorption cross sections, which are unique for each trace gas, acting as
their "fingerprints" and therefore enabling their specific detection.

For MAXDOAS, $I_0$ stands for the solar spectrum (known as the Fraunhofer

spectrum, with no Earth atmospheric absorptions), while $I$ represents the recorded
ground-based spectrum, which includes all the absorption and scattering processes.
However, and since the actual photon path is difficult to determine with accuracy (see
Sect. 4), the MAXDOAS calculations are done using relative absorptions between two
different optical paths: a zenith spectrum -that contains less absorptions and is
assumed as a reference spectrum- and other spectrum pointing to a given elevation
angle. Therefore, the direct product of the method is the differential slant column
density (DSCD), which can be defined as the difference in the integrated concentration
of a given absorber between the two selected pointing directions (more details about
the numerical procedure that lies behind can be found in Honninger et al., 2004, Plane
and Saiz-Lopez, 2006 and Platt and Stutz, 2008). Finally, these DSCDs are used as the
main input for the profile retrieval algorithms, which simulate the state of the
atmosphere with the purpose of reproducing the measured DSCDs. This final step
yields the optimal vertical concentration profiles.

**3 Experimental**

Briefly, MAXDOAS-1D instruments consist of a light collector attached to a
stepper motor that scans the atmosphere at different Viewing Elevation Angles (VEA,
see Fig. 1). The main feature added to the MAXDOAS-2D instrument is an additional
stepper motor for the azimuthal movement, hence allowing the light collector to freely
point to any angular direction in the atmosphere. This allows the evaluation of trace
gases absorptions for different Viewing Azimuth Angles (VAAs) (Fig. 1).

**3.1 MAXDOAS-2D description**

A MAXDOAS-2D instrument (Fig. 2) was built by the Atmospheric Chemistry and
Climate group at the Institute of Physical Chemistry Rocasolano (IQFRCSIC). Its main
elements are based on our previous MAXDOAS-1D instrument: a light collector
attached to a stepper motor, along with a focusing lens (80 mm focal length) are
responsible for collecting the scattered sunlight. An Ocean Optics, SMA 905 optical
fiber of 1-meter length conducts the light through an Ocean Optics, HR4000
spectrometer (which incorporates a linear silicon CCD array as detector). The
spectrometer wavelength ranges roughly from 300 nm to 500 nm and offers an
estimated spectral resolution (full width at half maximum) of about 0.5 nm. An
additional stepper motor was included for azimuthal movement. The instrument
incorporates all its components in an outdoor unit. Therefore, to maintain the
spectrometer temperature as steady as possible -for both mechanical and wavelength
calibration purposes- a Peltier cell was included. Additionally, an UPS device provides
the power supply and eliminates possible strong power peaks. Two webcams take
pictures of the cloud cover at each VAA, and monitor the instrument itself. The
instrument is autonomous and it runs on a homemade Java software. This software
controls the movement, the spectra collection and recording, the surrounding
accessories and automatically keeps it continuously measuring as long as the Sun is
over the horizon.

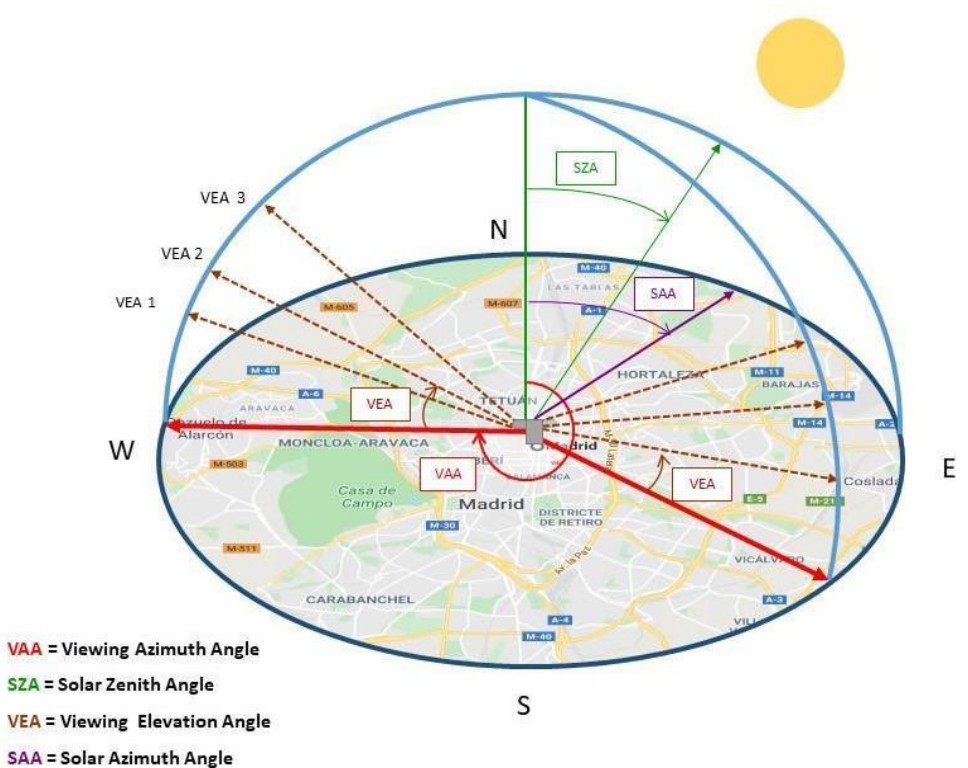


**Figure 1**. MAXDOAS-2D geometry diagram, the background of this picture represents
the Madrid city center taken from Google Maps.

**3.2 Location**

The MAXDOAS-2D instrument is located at the main campus of the Spanish
National Research Council (CSIC) in Madrid, Spain. It is placed on the roof of the
Instituto de Ciencias Agrarias (ICA) at a latitude of 40.4419° N and a longitude of
3.6875° W. The height of the building is approximately 70 m above ground level. This
location in downtown Madrid can be classified as an urban site, with the usual weather
of continental areas at mid-latitudes (i.e. hot and dry summers and cold winters), with
prevalence of clear sky days during the year. $NO_2$ typically presents strong spatial
concentration gradients in urban areas and traffic hot-spots have been reported in
Madrid (Borge et al., 2016). This makes it difficult to clearly predict how $NO_2$ will be
distributed, i.e., there is not a clear azimuthal direction preference for higher $NO_2$ at a
certain time. However, mesoscale simulations suggest that higher $NO_2$ mixing ratios
can be expected in the southern part of Madrid, considering population distribution
and commuting patterns (Picornell et al., 2019).

Due to some obstacles that blocked a clear view in some of the VAAs, a small
aluminum tower was built to overcome the viewing obstacles and the MAXDOAS-2D
instrument was fixed on top of it (see Fig. 2). Once the instrument was set up, we
aligned it for both angular movements -azimuthal and zenithal- with respect to the
geographical north and the local horizontal (i.e. perpendicular to the gravitational
plumb), respectively. This process was performed in two steps: first, the light collector
was coarsely oriented using levels and a compass. Then, the alignment was refined
doing a vertical scan of the Sun (which has a very well-known position vector) and its
angular surroundings at several different times of a clear sky day. The angular
differences between the measurements and the center of intensity of the registered
spectra (a similar approach was done in Ortega et al., 2015) were estimated and the
associated correction applied to the instrument.


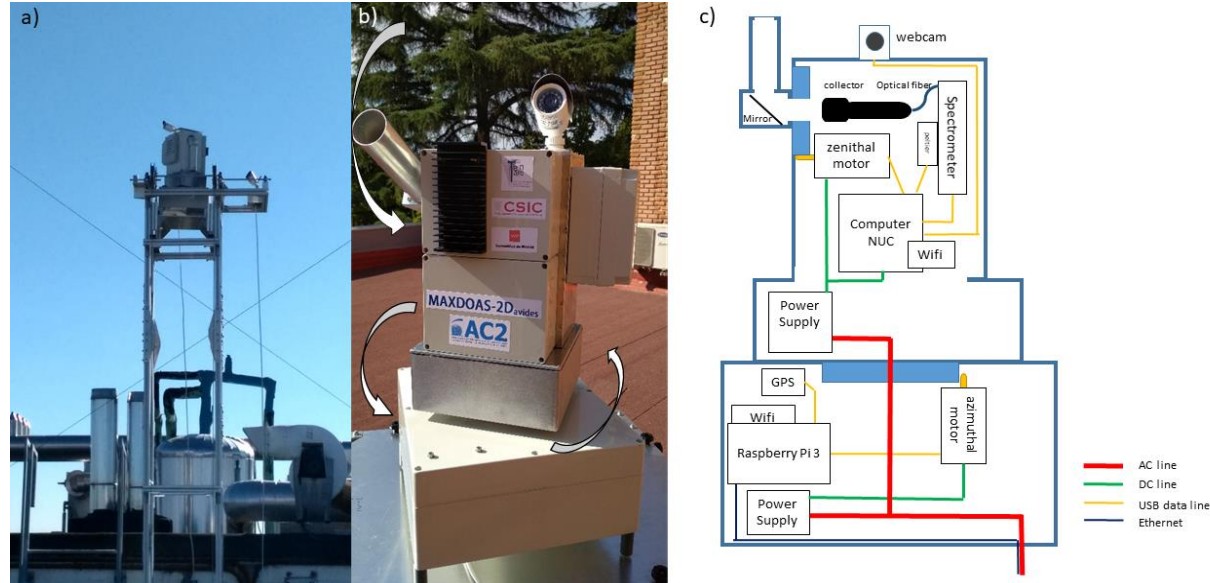

**Figure 2**. a) Aluminum tower with the instrument installed on top of it; b) MAXDOAS-
2D instrument; c) MAXDOAS-2D scheme.

**3.3 Measurements set up**

In order to sample and analyze a representative portion of the atmosphere over
Madrid, selected angular directions were chosen. Starting at a VAA of 0º (pointing to
the north), the MAXDOAS-2D rotated clockwise using steps of 20º in azimuth. In each
azimuth direction, the ensuing VEA vector was used: 1, 2, 3, 5, 10, 30 and 90 degrees.
Therefore, an entire azimuthal lap was completed when the light collector was back
again at VAA of 0 degrees.

For every measured spectrum, the spectrometer was able to correct for both
electronic offset and dark current effects. Other important parameters for the
measurements such as the integration time and the number of scans taken in each
angular direction were automatically calculated. More specifically, for this study we
set the goal of completing an azimuthal lap in approximately one hour (mainly for an
easier interpretation of the results and for the subsequent comparison with in-situ
instruments of Madrid´s air quality monitoring network). Hence, we chose 24 seconds
as the maximum exposure time in each angular combination.

The main advantage of this set-up is that we can observe the daily $NO_2$
variability over the entire city with a moderate temporal resolution (1-hour). The main
disadvantage is that observations for each VAAs averaged over such a short integration
period may be affected by microscale phenomena. Nonetheless, $NO_2$ concentration
gradients are particularly strong in space (Borge et al., 2016). Therefore, this time
resolution may be well suited to characterize both the azimuthal and the horizontal
gradients of $NO_2$.

**4 Analysis methods**

The absorptions of the molecular oxygen dimer ($O_4$) and $NO_2$ were measured
for the entire campaign and for two spectral windows: 352-387 nm (UV region) and
438-487 nm (VIS region). The analysis settings applied for the UV and VIS regions are
summarized in Tables 1 and 2, respectively. These configurations mainly follow those
used in Wagner et al., 2019.

**Table 1.** DOAS spectral settings for the retrieval of $O_4$ and $NO_2$ in the UV.

| Parameter | Value |
| --- | --- |
| Fitting window | 352-387 nm |

| | |
|---|---|
| Wavelength calibration | Based on reference solar atlas (Chance and Kurucz, 2010) |
| Zenith reference | Scan |
| Polynomial Order | 5 |
| Intensity Offset | Order 2 |
| Shift | The measured spectra and Ring were allowed to shift and stretch (order 1) in wavelength. |

| Molecule | Cross section |
|---|---|
| $O_4$ | 293 K (Thalman and Volkamer, 2013) |
| $NO_2$ | 298 K (Vandaele et al., 1998) |
| $O_3$ a | 273 K (Serdyuchenko et al., 2014) |
| $O_3$ b | 223 K (Serdyuchenko et al., 2014) |
| HCHO | 297 K (Meller and Moortgat, 2000) |
| HONO | 296 K (Stutz et al., 2000) |
| Ring_a | Calculated by QDOAS |
| Ring_b | Ring_a spectrum multiplied by $\lambda^{-4}$ |


**Table 2.** DOAS spectral settings for the retrieval of $O_4$ and $NO_2$ in the VIS.

| Parameter | Value |
| --- | --- |
| Fitting window | 438-487 nm |
| Wavelength calibration | Based on reference solar atlas (Chance and Kurucz, 2010) |
| Zenith reference | Scan |
| Polynomial order | 5 |
| Intensity offset | Order 2 |
| Shift | The measured spectra and Ring were allowed to shift and stretch (order 1) in wavelength. |

| Molecule | Cross section |
| --- | --- |
| $O_4$ | 293 K (Thalman and Volkamer, 2013) |
| $NO_2$ | 298 K (Vandaele et al., 1998) |
| $O_3$ a | 273 K (Serdyuchenko et al., 2014) |
| $O_3$ b | 223 K (Serdyuchenko et al., 2014) |
| $H_2O$ | 296 K (Rothman et al., 2010) |
| Glyoxal | 296 K (Volkamer et al., 2005) |
| Ring_a | Calculated by QDOAS |


The selected differential absorption cross sections -along with the spectral
window and parameters included in Tables 1 and 2- were adjusted to the measured
differential optical density using the QDOAS spectral fitting software (http://uv-
vis.aeronomie.be/software/QDOAS/). Figure 3 shows examples of spectral detection
of $O_4$ and $NO_2$ for both the UV and VIS regions.

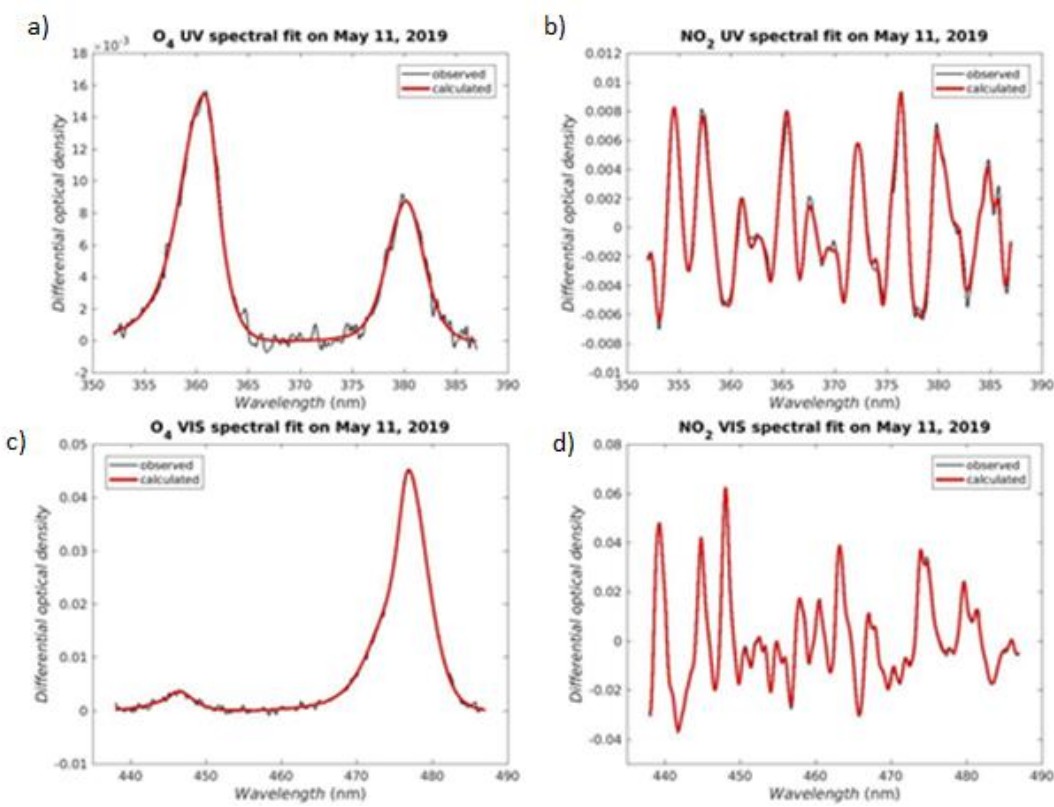


**Figure 3**. Spectral detection of $O_4$ (a) and (c) and $NO_2$ (b) and (d), red lines represent
the calculated optical densities and black lines are the measured optical densities. The
viewing geometry of each detection was: (a) VEA 2° and SZA 26.4°; (b) VEA 3° and SZA
24.7°; (c) VEA 1° and SZA 22.6° and (d) VEA 1° and SZA 48.7°.

**4.1 Cloud-screening and quality filtering**

The algorithms for MAXDOAS retrievals of trace gas vertical profiles are based on estimating the light paths (along with their corresponding scattering probability). A significant cloud cover could noticeably impact the calculations, mainly because of multiple scattering effects, adding large uncertainties to the retrieval process. For this reason, the set of measured spectra was cloud-screened using the cloud-free AERosol RObotic NETwork (AERONET) database. The AERONET databases are reported with three quality levels, in particular, we used the Level 2.0 (cloud-screened and quality-assured) database provided by the AERONET instrument in Madrid. This information is combined with the photos taken by the camera installed on the MAXDOAS. As mentioned in Sect. 3.1, this webcam points at the same azimuthal direction as the light collector. We estimated the cloud cover using a code that gets the RGB coordinates - the three chromatists of the blue, green and red- and it changes them into LCh coordinates -L indicates lightness, C represents chroma and h is the hue angle. Based on criteria of luminosity, colour and saturation, the code estimates the cloud index (percentage of estimated cloud cover in a given azimuthal sky view). Filtering out cloudy skies with precision is rather challenging, therefore we established a threshold to get as many clear sky views as possible. In order to do that, we first crossed our measured DSCDs with the AERONET 2.0 database, and subsequently, using the photos, we discarded the cycles taken with an estimated cloud index higher than 40 %.

In addition to cloud screening, several other quality filters were applied to the DSCDs: firstly, every DSCD that yielded either a relative uncertainty larger than 1 or a residual root mean square (RMS) higher than 0.01 (in optical density units) was rejected. After that, we estimated the DSCDs detection limit for a given trace gas as the ratio of the residual RMS (in optical density units) associated to each DSCD and the maximum value of the differential cross section of that trace gas. Then, we discarded the DSCDs that had an absolute value lower than twice the derived detection limit (a

similar approach was carried out in Peters et al., 2012). Finally, we used the daily plus/minus three standard deviation criterion that AERONET applies for its cloud-filtered data, keeping the DSCD that falls within plus/minus three standard deviations from each daily mean.

**4.2 Inversion algorithm and vertical profiles**

An inversion algorithm method is applied to the measured DSCDs to estimate the light paths and subsequently derive the trace gas vertical concentration profile. For this work we have used the bePRO inversion algorithm (Clémer et al., 2010). The original calculation was built based on the Optimal Estimation Method (OEM; Rodgers, 2000) and it comprises two steps: first, the light paths and the vertical profiles of irradiance extinction are calculated using the $O_4$ DSCDs; then, the target trace gas vertical concentration profile is retrieved using the corresponding light paths and measured absorption. In order to do that, bePRO simulates the atmospheric state characterizing several different physical phenomena including pressure and temperature vertical profiles, Rayleigh and Mie scattering events (along with their respective phase functions), the effect of the surface albedo, the light path geometries or the irradiance extinction processes. Once the atmospheric vector state is defined, its combination with a certain vertical concentration profile results in the simulated DSCDs. This vertical profile is iterated until the generated set of simulated DSCDs is optimized with respect to the measured DSCDs so that the residual is minimized. As a result, an optimal vertical profile is obtained when the iteration is finished for each MAXDOAS cycle.

The measured $O_4$ DSCDs are used to estimate the light paths for each VEA since they are related to the square of the atmospheric $O_2$ profiles, which are well-known.

This profile is fairly steady during the day and does not heavily depend on chemistry
factors. Therefore, the measured $O_4$ DSCDs can provide information on the irradiance
extinction in the atmosphere. This extinction profile is usually associated with the
aerosol extinction coefficients and thus, its vertical integration yields the aerosol
optical depth (AOD). These aerosol extinction profiles are required to subsequently
evaluate trace gas profiles since they strongly affect the relative light paths and hence
the concentration profiles derived from them.

Once the light paths are computed, and with the purpose of best simulating the

measured DSCDs, a linear analysis process is performed for the measured DSCDs of
the target trace gas, yielding the optimal vertical concentration profile. The vertical
integration of this concentration profile is called the vertical column density (VCD).

The retrieval consists of an iterative, nonlinear system of equations, and hence

there is no unique solution. This means that an a priori profile is needed, both for
starting the iterations and to avoid the final solution to be non-realistic (i.e. with no
physical meaning). In order to construct these a priori profiles we used exponentially
decreasing curves as follows:

$$ap\,(z) = \frac{VC_i}{sh} exp\left(\frac{-z}{sh}\right)$$                    (2)


where $ap\,(z)$ is the a priori vertical profile at a certain altitude $z$, $VC_i$ is the

vertical integration of the profile for the MAXDOAS cycle *i* and *sh* is the scaling height
constant. We used 0.5 km as the scaling height constant for all the a priori profiles
(Hendrick et al., 2014). Regarding the VC, we assumed an AOD of 0.05 for the $O_4$
retrieval, while for $NO_2$ we applied the geometrical approximation followed in
Hönninger et al., 2004, taking the measured DSCD at VEA 30° for every MAXDOAS
cycle. This approximation assumes that most of the absorption events are located
below the scattering height.

With respect to the remaining atmospheric parameters, we chose typical values

for urban environments: surface albedo of 0.07, correlation length of 0.4 km and an a
priori covariance factor of 1 (see Hendrick et al., 2014). Concerning the vertical grid of
the retrievals, we chose the following layers: from the ground to 8 km height we used
layers of 200 m thickness. Then, we divided the remaining height, up to 90 km, in levels
of 2 km thickness each. We use the air number density vertical profile since it is directly
related to the number of $O_4$ absorptions, and therefore to the $O_4$ DSCDs. Hence the
relative differences, particularly for lower VEAs, between the measured and simulated
$O_4$ DSCDs are usually assigned to aerosol extinction. Note however, as shown below,
that uncertainties in the air number density profiles -arising from uncertainties in the
values or shape of the temperature and pressure profiles- could also contribute to such
differences (Fig. 4).

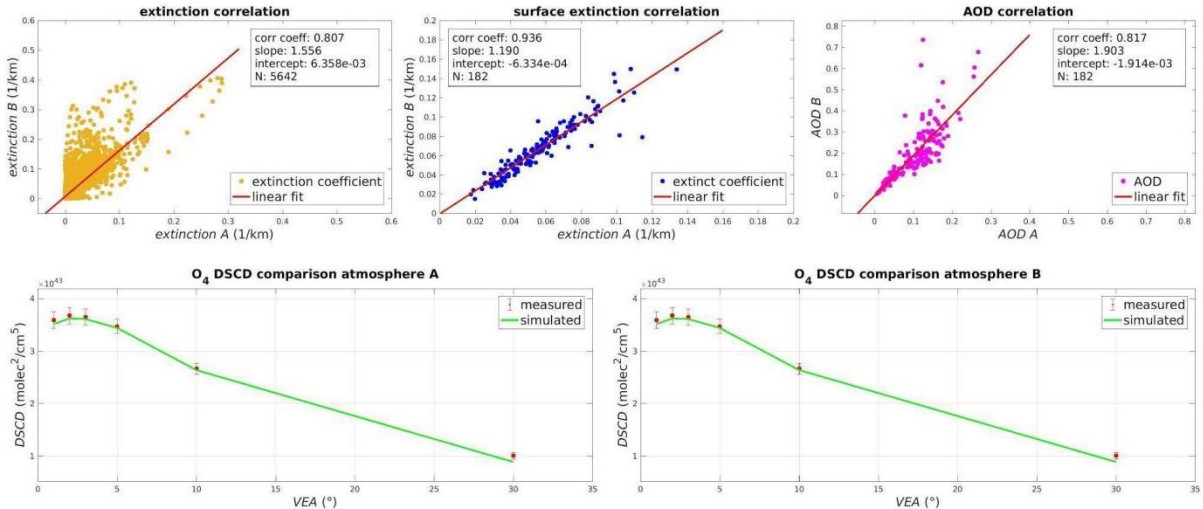


**Figure 4**. Comparison of retrieved aerosols using two different atmospheric
profiles: the US Standard (atmosphere A) and the US Standard adapted to the altitude
above sea level of Madrid (atmosphere B). The comparison was carried out for a clear
sky day (May 11, 2019).

Here we compare the simulation of $O_4$ DSCDs using two different sets of
atmospheric profiles: i) the US Standard, and ii) the same profile but interpolating the
pressure profile to Madrid´s height above sea level (mean value of 667 m). This means
that the temperature profile is assumed to be the same but the pressure profile is
shifted less than 10%, so there are no major variations within the profiles. The lower
row in Fig. 4 shows that both atmospheric profiles result in almost the same set of
simulated $O_4$ DSCDs, however the aerosol extinction coefficients differ significantly
(although less for the surface layer coefficients, defined as the extinction coefficients
within the ground layer, between 0 and 200 m height), and consequently, the AOD
also varies. From this we infer that:

i)    the retrieval is mainly driven by the measured DSCDs, which leaves a
relatively low weight for the chosen atmospheric profiles (pressure and
temperature). Therefore, we can obtain consistent correlations between
the measured and simulated $O_4$ DSCDs.


ii)   we cannot reliably assign the extinction coefficients at each layer to
aerosols (especially for atmospheric layers above the surface layer), but
rather consider them as irradiance extinction coefficients.


Furthermore, we have assessed the impact of the pressure and temperature
profiles choice on the trace gas retrieval. As can be noted in Fig. 5, there is no
significant effect coming from this choice on the simulated $NO_2$ DSCDs. These are
basically the same (and with very good agreement with the measured DSCDs), as well
as the derived concentration coefficients and their integration (VCD).

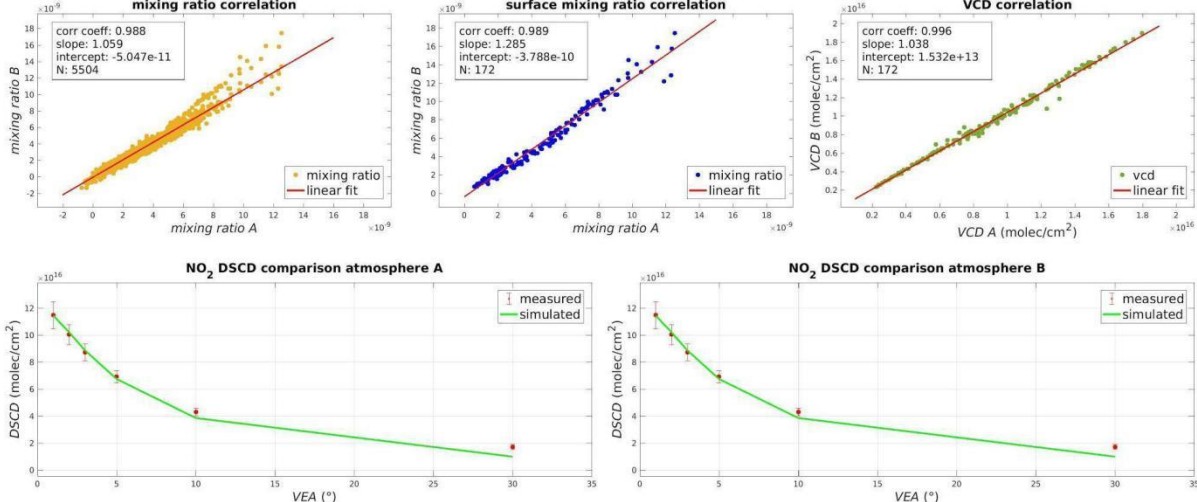


**Figure 5**. NO$_2$ retrieval comparison using two different atmospheric profiles: the US
Standard (atmosphere A) and the US Standard adapted to the altitude above sea level
of Madrid (atmosphere B). The comparison was carried out for a clear sky day (May
11, 2019), considered as representative of the measurement period.

We further evaluated if a similar behavior can be expected for larger variations
in the pressure and temperature profiles. We first obtained the average surface
temperature and pressure values during the campaign (May-July, 2019). With the
inclusion of these values in the retrieval, we found that, within the first 10 km height,
the RMS of the relative variations with respect to the standard atmosphere were about
8 %. Although it is a small change, it is indeed not negligible. Nonetheless, when
evaluating light paths, the relative changes were below 2%. Therefore, here we use
the US Standard atmospheric profiles for the NO$_2$ retrievals.

Table 3 summarizes the average uncertainties (using one standard deviation for

each component) of the retrieval, along with their relative contributions, for the
ground layer (0-200 m height). The mean, overall uncertainty for $NO_2$ in both spectral
regions is in the order of 10%.

**Table 3.** Summary of average uncertainties of the retrieval in both spectral regions.

| Variable \ Trace gas | $NO_2$ UV (%) | $NO_2$ VIS (%) |
|:---:|:---:|:---:|
| **Irradiance Extinction** | 7.7 | 5.1 |
| **DSCD** | 4.8 | 3.2 |
| **Surface Mixing Ratio** | 5.0 | 8.7 |
| **Total** | 10 | 11 |


**4.3 Estimation of $NO_2$ horizontal gradients**

Making use of the different paths that photons travel through the atmosphere

for different wavelengths, we can estimate the horizontal distribution of $NO_2$. We use
the estimated horizontal light paths at two wavelengths, 360.8 nm and 477 nm, for
the surface layer (0-200 m height). The different light paths at 360.8 and 477 nm
provide information about the horizontal distribution of $NO_2$ mixing ratios within the
surface layer. In order to evaluate these horizontal paths, we have a code that
implement the RTM equations based on previous pioneering work (Solomon et al.,
1987). These equations yield a vector of scattering events along with their respective
probabilities. If we take a VEA of 0 degrees (i.e. horizontal viewing), then the scalar
product of such vectors produces the length of the horizontal light path.

We computed this for every MAXDOAS cycle and for both wavelengths, yielding

typical -representative- horizontal distances of about 8-10 km for the UV (at 360.8 nm)
and between 15-20 km for the VIS window (at 477 nm). The next step follows the
"onion-peeling" approach proposed by Ortega et al. 2015 (the strong dependence of
scattering with wavelength means that shorter wavelengths result in shorter light
paths). We assign the UV (i.e. 360.8 nm) mixing ratios ($mr_{uv}$) and their expected
horizontal paths ($d_{uv}$) to the first peel ($mr_A$, meaning zone A). Then the second peel
(zone B, $mr_B$) can be derived as:

$$mr_B = \frac{mr_{vis} \times d_{vis} - mr_{uv} \times d_{uv}}{d_{vis}} \tag{3}$$


Thereby, deriving mixing ratios ($mr_a$ and $mr_b$) representative of two different

horizontal distances for each VAA.

**5 Results**

**5.1 $O_4$ and $NO_2$ DSCDs assessment**

An estimation of the overall goodness of the profile retrieval comes from the
correlation between the measured and simulated DSCDs for the entire campaign (Fig.
6). The fit between the measured and the simulated DSCDs shows correlations ($r^2$) very
close to 1 for both $O_4$ and $NO_2$ in the UV and VIS regions.

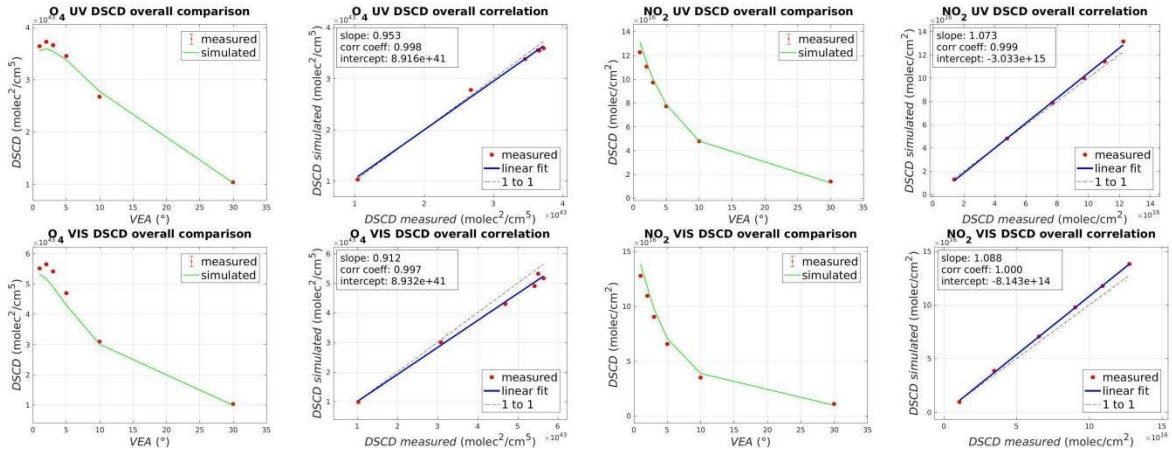


**Figure 6**. Comparison between simulated and measured DSCDs of O₄ and NO₂. Red dots represent the measured DSCDs for each VEA, averaged over the entire campaign.

**5.2 Two-dimensional maps**

We now combine the VAA and height for each azimuthal cycle of the MAXDOAS-2D to generate a two-dimensional concentration map. Fig. 7 shows an example of the O₄ retrieval in the UV for a given azimuthal cycle. Fig. 7 also shows the comparison and correlation of measured and simulated DSCDs for that azimuthal cycle, along with the evolution of retrieved AOD. The AOD varies between 0.05 and 0.18 within this azimuthal cycle (Fig. 7, upper panel). The contour plot shows the irradiance extinction coefficient profiles with maximum values of 0.14 km⁻¹ (near the ground and at around 40º VAA) associated with aerosol extinction (see discussion in Sect. 4.2). Note the enhanced extinction at about 2 km height pointing at 50 VAA. This could be due to uplift of particulate matter emitted by traffic (there is a main road at that location) (Carnerero et al., 2018). Further research is needed to better establish the vertical distribution of aerosols in Madrid, and their diurnal evolution.

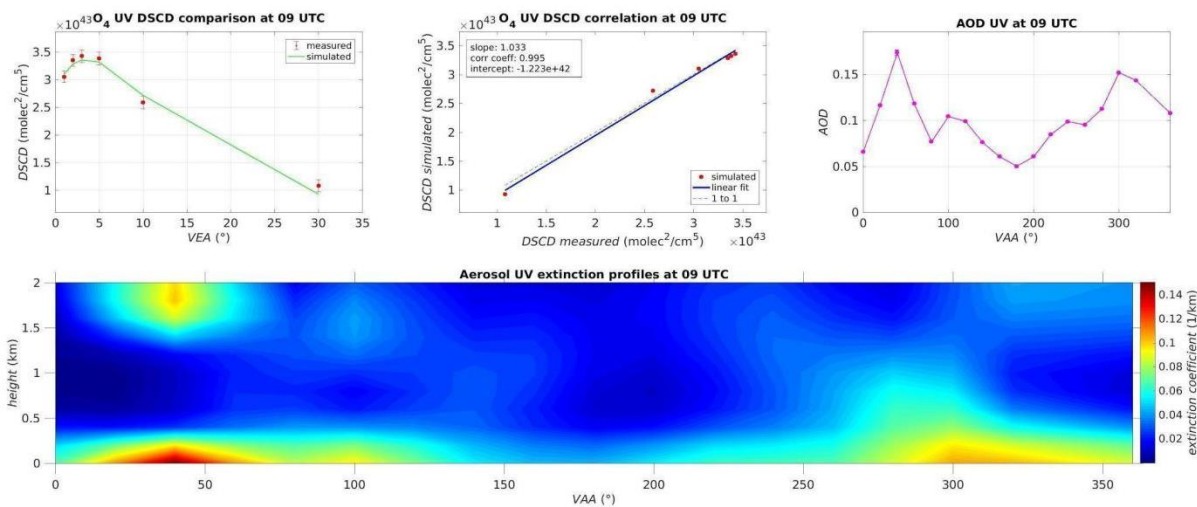


        **Figure 7**. Example of $O_4$ and AOD retrievals in the UV region at 9 UTC on May

11, 2019. These contour plots are smoothed from adjacent VAA data points separated
by 20º in order to estimate the azimuthal distribution of the irradiance extinction
coefficients over Madrid.

Figure 8 shows a two-dimensional representation of $NO_2$ on May 11, 2019 at

two different hours (6 UTC and 12 UTC, respectively). Both contour plots show
maximum $NO_2$ values of 12 ppbv at 6 UTC and 8 ppbv at 12 UTC, when the instrument
is pointing south (i.e. VAA of 180º). We chose to show this day as an example since it
was a clear sky day and yielded $NO_2$ mixing ratios that were representative of the
period of measurements. These values correspond to the layer near the ground and
are in good agreement with our previous MAXDOAS observations in Madrid (Garcia-
Nieto et al., 2018). The retrieved azimuthal distribution of $NO_2$ agrees with previous
reports that show higher pollution levels in the southern part of Madrid (Picornell et
al., 2019). $NO_2$ VCDs range from $5 \times 10^{15}$ molecules cm$^{-2}$ (at 12 UTC and pointing at 300
° VAA) up to $15 \times 10^{15}$ molecules cm$^{-2}$ (at 12 UTC and pointing at 200º VAA), with an
average value of $1 \times 10^{16}$ molecules cm$^{-2}$. Although there are different $NO_X$ emission
rates at both times of the day (6 and 12 UTC), the increase in the boundary layer height
(de la Paz et al., 2016) during the day contribute to the similar values of VCDs at both
hours but generally lower surface mixing ratios at 12 UTC.

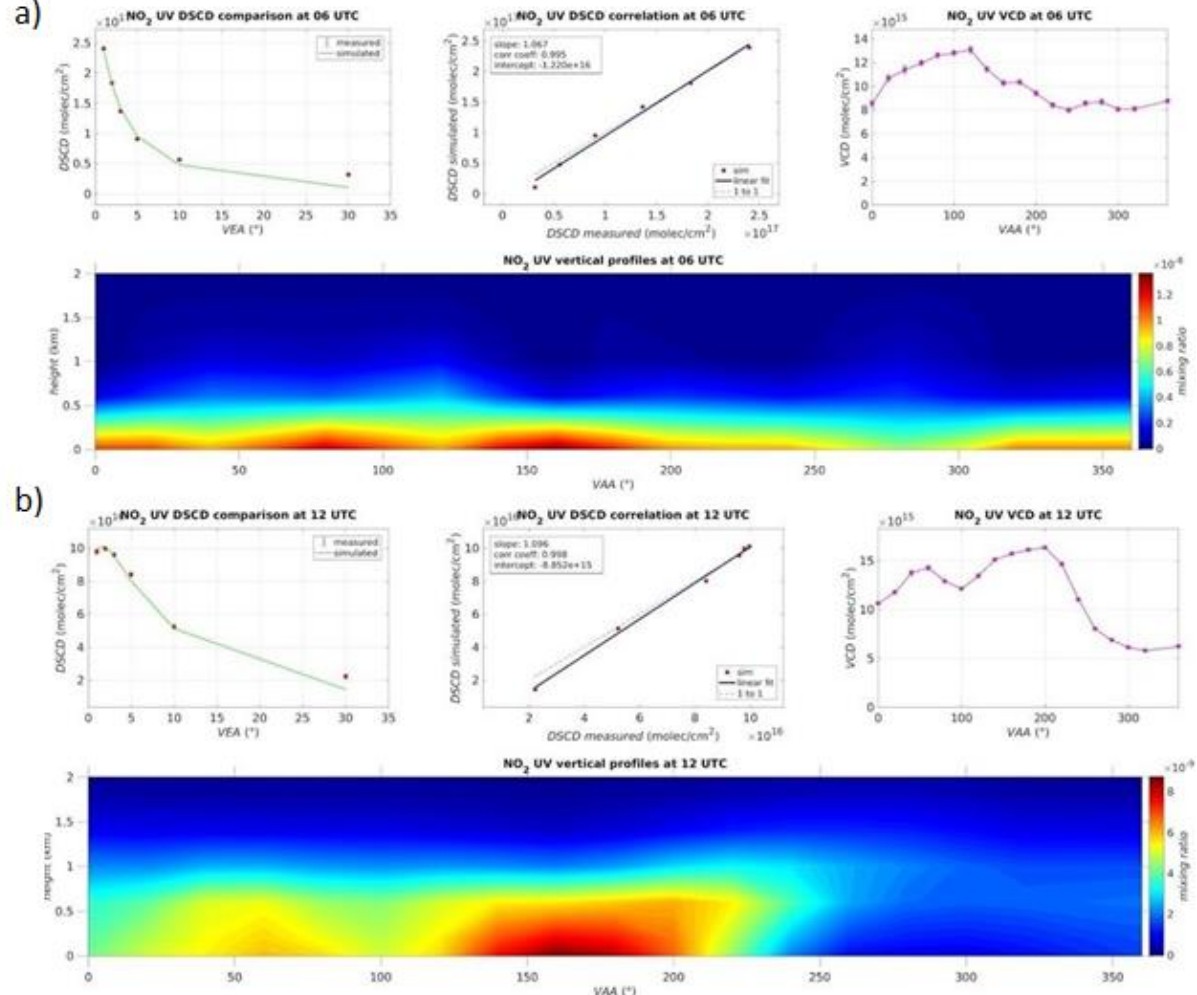


**Figure 8**. NO$_2$ vertical distribution retrieved in the UV region at 6 UTC (a) and at 12 UTC
(b) on May 11, 2019. These contour plots are smoothed from adjacent VAA data points
separated by 20º in order to estimate the azimuthal distribution of NO$_2$ over Madrid.

We have also analyzed HONO DSCDs using the same DOAS analysis

configuration as in Garcia-Nieto et. al., 2018. Figure 9 shows a two-dimensional
representation of HONO on May 11, 2019 at 6 UTC. We retrieve surface layer peak
values of 0.7 ppbv pointing at 50° of VAA in the early morning, in agreement with
previous studies for HONO in urban environments (see Hendrick et al., 2014; Ryan et
al., 2018). The VCDs at 6 UTC range from $6 \times 10^{14}$ to $1.2 \times 10^{15}$ molecule cm$^{-2}$.

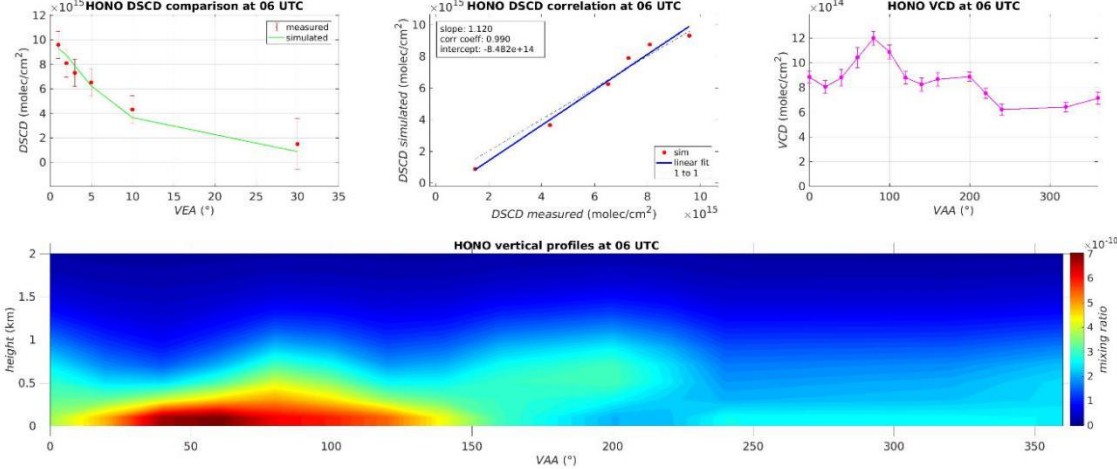


**Figure 9.** HONO vertical distribution retrieved in the UV region at 6 UTC. These contour
plots are smoothed from adjacent VAA data points separated by 20° in order to
estimate the azimuthal distribution of HONO over Madrid.

**5.3 Horizontal distribution of NO$_2$**

Based on Eq. (3), we derive the horizontal distribution of NO$_2$ in the surface
layer (0-200 m height) using the measured NO$_2$ DSCDs at a VEA of 1° (which can safely
be regarded as almost horizontal viewing, since its mean scattering height typically
falls below 200 m height, i.e. within our ground layer). Figure 10 shows an example of
surface layer NO$_2$ mixing ratios over two radial distances horizontally measured from
the MAXDOAS-2D instrument (using the UV and the VIS NO$_2$, respectively, as explained
in Section 4.3), located at the center of the plot. The highest mixing ratios occur during
the first sunlit hours (7-8 UTC), coincident with the morning peak of NO$_X$ emissions in
Madrid (Quassdorff et al., 2016). This early morning peak is followed by a gradual
decrease in surface layer NO$_2$ mixing ratios during the day. Note that NO$_2$ is
predominantly located in the southern part of the semisphere (VAA from 90º to 270º).
In follow up work we will combine the horizontal distribution of $NO_2$ with a chemical
transport model to further understand $NO_2$ transport dynamics throughout the day.

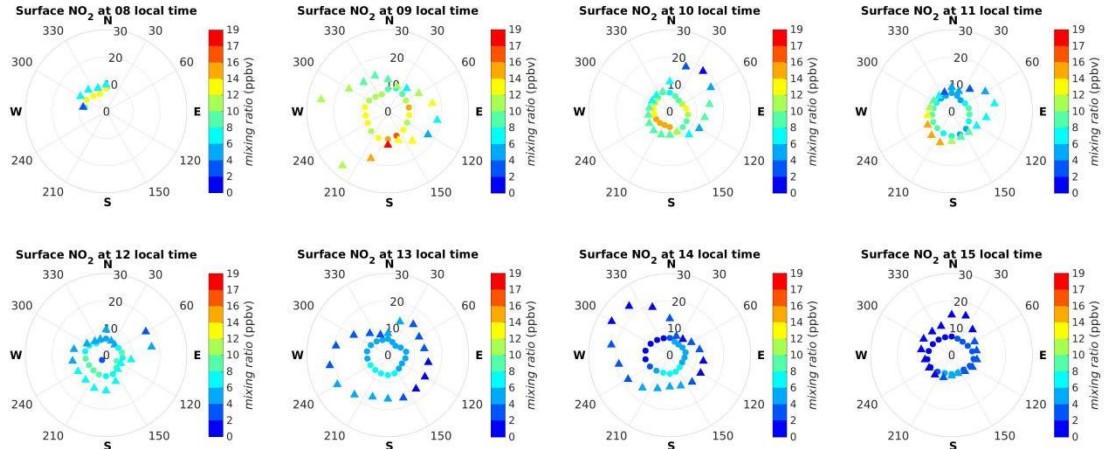


**Figure 10.** Polar plots of $NO_2$ within the surface layer (0-200 m height) for May 11,
2019. Please note that these polar plots extend over a direction perpendicular to those
shown in Fig. 8. Here, circles are used for the UV (shorter horizontal light path) and
triangles for the VIS (larger horizontal light path). Both symbols stand for the mean
horizontal light path within the surface layer at each spectral region.

**5.4 Correlation with Madrid´s in-situ air quality monitoring stations**

We suggest that MAXDOAS-2D mesoscale observations may complement the
information provided by the local air quality monitoring network based on reference
analytical techniques (according to Directive 2008/50/EC). While air quality monitors
of the reference network provide information about ambient concentrations in their
specific locations (currently 24 air quality monitoring stations measure $NO_2$ within the
city, see AM, 2019), MAXDOAS-2D observations produce near ground-level
concentrations averaged over the optical path in a given direction. That prevents us
from quantitatively comparing both types of observations. Nonetheless, we analyzed
their correspondence using the $NO_2$ concentrations measured by the in-situ
instruments throughout the entire city, and the $NO_2$ mixing ratios within the surface
layer derived from our MAXDOAS-2D instrument over the 2-month period (May-June,
2019). For this comparison, we considered the air quality monitoring stations within a
distance from the MAXDOAS-2D equal or lower than 10 km (thus remaining 20 air
quality monitoring stations). Since this is the typical horizontal light path for the UV
region, we decided to include only the $NO_2$ values retrieved in the UV region for the
comparison. Strong gradients between the values measured by the in-situ instruments
are typical.  Therefore, and considering that we are mainly interested in their temporal
correlation with respect to our measurements, we compare both the in-situ $NO_2$ and
surface layer MAXDOAS-2D hourly-averaged data. Note that for the MAXDOAS-2D,
this approximately corresponds to averaging the surface layer values for each
azimuthal lap, given that each azimuthal lap takes approximately 1 hour to complete.

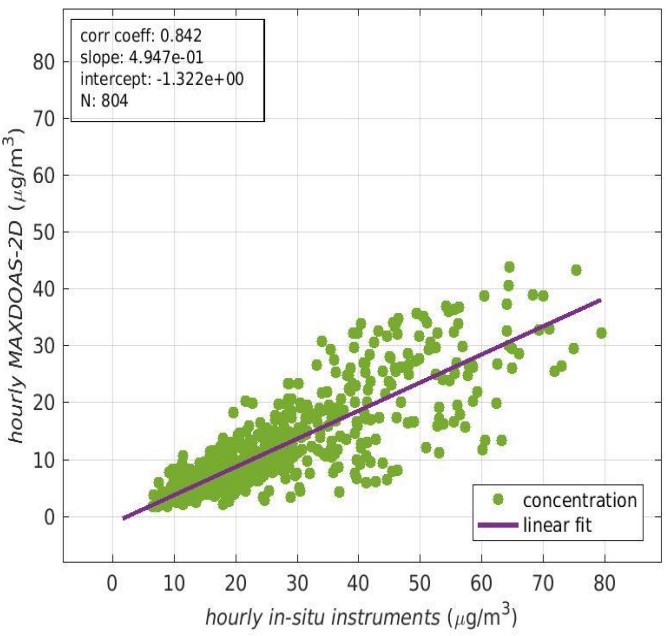


**Figure 11**. Correlation between in-situ observations from Madrid´s air quality
monitoring network and those derived from the MAXDOAS-2D instrument for the
surface layer (0-200 m height).

Despite the different spatial representativeness, Figure 11 shows a reasonably

good correlation coefficient of 0.842 between both datasets for the two-month
campaign. The slope is lower than 1, this can be explained by the typical $NO_2$ vertical
profiles in urban environments. Simulations performed over Madrid with a high-
resolution Eulerian air quality model (Borge et al., 2018) yielded an exponentially
decreasing with height $NO_2$ profile. Therefore, the MAXDOAS-2D mixing ratios, which
represent an average across the surface layer (0-200 m height), are not expected to
quantitatively match the values of in-situ instruments, located close to the surface
(between 0-10 m height). Similar conclusions -and slopes comparable to the one
retrieved above- regarding the correlation between in-situ and MAXDOAS instruments
can be found in previous works (Schreier et al., 2020; Kramer et al., 2008; Chan et al.,
2020). In addition, there is a good temporal correlation between in-situ and
MAXDOAS-2D measurements over an extended period of time.

**6 Summary and Conclusions**

An analysis of $O_4$, $NO_2$ and HONO vertical concentration profiles in the urban

atmosphere of Madrid (Spain) has been performed over two months (from May 6 to
July 5, 2019). We analyzed the absorptions and derived the corresponding DSCDs for
both trace gases in the UV and VIS regions. Then, the corresponding profiles were
retrieved using a RTM. In this step, we assessed the impact of different atmospheric
profiles (pressure and temperature) in the retrieval results, and found that the set of
chosen atmospheric profiles has a small impact on the $O_4$ retrieval and the estimation
of light paths. However, there is a noticeable change in the irradiance extinction
profiles, which makes difficult to quantitatively assign extinction due to aerosols,
especially in heights above the boundary layer.
The overall comparison of measured and simulated trace gas DSCDs showed
that they were in very good agreement (with correlation coefficients close to 1),
supporting the reliability of the observations. The MAXDOAS-2D instrument provides
the first two-dimensional view (in height and VAA) of pollution concentration in the
city of Madrid. Exploring one day (May 11, 2019) we compared two hours: the peak
rush hour and noon time, obtaining $NO_2$ maximum values of 12 ppbv and 8 ppbv
respectively, both maxima pointing to the south direction. Two-dimensional HONO
measurements were also made with mixing ratio peaks of 0.7 ppbv in the early
morning, and VCDs ranging from $6x10^{14}$ to $1.2x10^{15}$ molecule cm$^{-2}$.

We have also inferred information on the horizontal gradient of $NO_2$ within the
surface layer making use of the strong dependence between wavelengths and light
paths across the $NO_2$ absorption spectrum. The resulting "onion-peeling" figures
indicate peak values of $NO_2$ in the early morning and in the southern section of the city
(around 180 º VAA), it resulted in a gradual decrease in $NO_2$ mixing ratios during the
day, maximum values of $NO_2$ appear in the southern part of the semisphere. Finally,
we suggest that the new mesoscale information provided by the MAXDOAS-2D
instrument helps in the study of pollution transport dynamics. MAXDOAS-2D and in-
situ instruments provide different information, and thus, combining both can improve
our understanding of the complex issue of air pollution in the city of Madrid.

**Author Contribution**

A.S-L. devised the research. D.G-N. and N.B. carried out the measurements and
analyzed the data. D.G-N., N.B., R.G. and A.S-L. analyzed and interpreted the results.
D.G-N. wrote the manuscript with contributions from all co-authors.

**Acknowledgements**

The authors want to thank Manuel Perez and David Armenteros for technical
assistance with the instrument, and David de la Paz for model assistance. This work
was supported by the TECNAIRE project ("Técnicas innovadoras para la evaluación y
mejora de la calidad del aire urbano") S2013/MAE-2972. We would also like to thank
Juan Ramón Moreta González (PI) and his staff for establishing and maintaining the
AERONET sites in Madrid used in this investigation. We acknowledge support of the
publication fee by the CSIC Open Access Publication Support Initiative through its Unit
of Information Resources for Research (URICI)

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
