# Peer review of "Two-dimensional monitoring of air pollution in Madrid using a MAXDOAS-2D"

_Atmospheric Measurement Techniques, 2020_

## Referee Comment (RC1) · Anonymous Referee #1 · 24 Jul 2020

The present manuscript presents a complete analysis of O4 and NO2 vertical profiles during three months in Madrid, Spain with the aid of ground-based MAX-DOAS 2-D observations. The aerosol and NO2 vertical profiles in multiple viewing azimuth directions are presented here as well as the horizontal NO2 distribution around the measurement site. Finally, the 2-D MAX-DOAS NO2 near-surface concentrations are compared with the in-situ NO2 measurements in Madrid.

I recommend the publication of the manuscript after consideration of a major number of specific comments:

Specific comments:

1. Page 1, Line 19: Please write the spatial resolution of the mesoscale events.
2. Page 1, Line 27: In my understanding, you used one inversion algorithm (not inversion algorithms) for the aerosol and the NO2. Please correct that and write the name of the inversion algorithm that is used (bePRO).
3. Page 1, Abstract: I would recommend that you write in a more clear way, the main findings of this study and the main contributions/innovations that you have made.
4. Page 2, Line 49: I would recommend to write that you have developed two MAX-DOAS instruments and not just MAX-DOAS instruments.
5. Introduction: It would be valuable to add a paragraph in which you cite previous MAX-DOAS studies of two-dimensional measurements (like Ortega, Schreier, Wang, Dimitropoulou etc.) as well as studies where MAX-DOAS observations are compared with in-situ measurements.
6. Section 3.2: Where do you expect to measure higher NO2 concentrations (North, South etc.)?
7. Page 7, Line 193:  In your study, one complete MAX-DOAS scan takes one hour. The advantage is that you have a very nice horizontal sampling but at the other hand, you risk to measure the same NO2 air mass in multiple azimuthal directions (for example, during one hour, the NO2 that you observe in the North can be moved by the wind in the North East direction). Please add a sentence in which, you make clear the advantages and disadvantages of your choice.
8. Page 11, Line 252: After the filtering of the MAX-DOAS measurements, which is the percentage of accepted scans?
9. Page 11, Line 264: The RTM is the forward model and the bePRO is the inversion algorithm. Please correct this.
10. Page 12, Line 290: It's not exactly an analogous process because for the O4 and aerosol, non-linear calculations are performed and for trace gases as NO2, we have linear calculations. Please verify if it's the case for bePRO and correct or not this sentence.
11. Page 13, line 310-318: You have used Standard atmosphere profiles, which are widely used in studies like the present one. But, you should include an uncertainty estimate of using a standard profile instead of a real profile (by meteorological measured data)
12. Section 4.2: You should a paragraph in which you present an average error estimate of the retrievals and add a Table with all the error sources (smoothing error etc).
13. Section 4.3: In your results, you should discuss the range of the estimated horizontal distances for the UV and Vis during your measurement period
14. Figure 6: These results are from which measurement day and scan/hour? I assume that it is not the whole period, right?
15. Figure 7: How do you explain the aerosol peak at around 50 deg. VAA and in high altitude?
16. Page 20, Line 465: Why do you use the UV distance and the Vis which is larger?

17. Figure 10: Please include a 1:1 line and put the same axis limits in both x, y axis in order to quantify rapidly the underestimation on the near-surface NO2 concentrations by the MAX-DOAS
18. Page 21, Line 480: You write that the slope is lower than 1 (it is 0.4) which is true but you should add a sentence in which you discuss this finding. Is it in agreement with previous studies that compared MAX-DOAS and in-situ?
19. Conclusions: You should make this section larger and discuss more your results
20. Through the whole manuscript, references should be added, as I mentioned in previous comments

Technical corrections

1. Page 2, line 34: gaseous pollutant concentrations instead of gaseous pollutants concentrations
2. Page 3, line 73: path lengths instead of paths lengths
3. Page 11, Line 256: inversion algorithm method instead of inversion algorithms

---

## Referee Comment (RC2) · Anonymous Referee #2 · 17 Aug 2020

In their manuscript "Two-dimensional monitoring of air pollution in Madrid, Spain using a MAXDOAS-2D instrument", the authors report on measurements in Madrid using a new MAX-DOAS instrument with both elevation and azimuth pointing capabilities. Examples of NO2 profile retrievals are discussed and some results of onion peeling retrievals presented. Finally, a comparison is performed between hourly mean values from the lowest MAX-DOAS profile level and data from the air quality network, showing good correlation.

The manuscript is generally clear and well written but lacks detail in many places. It also does not provide reference to the many existing studies using similar instruments,

performing similar retrievals, and addressing similar research questions.

My main problem with this manuscript is however the lack of novelty: In fact, I do not see anything new in this manuscript on instrument development, DOAS retrievals, profile retrievals, the onion peeling approach or the validation of the retrievals. The instrument is similar to many others operated (see Kreher et al., 2020), the DOAS retrieval is performed using the freely available software QDOAS, the profile retrieval is using the software BePro, the onion peeling follows the work by Ortega et al. and the validation is limited to a single figure showing measurements from a not further defined time period. I therefore unfortunately cannot recommend this manuscript for publication in Atmospheric Measurement Techniques.

The measurements of the 2d-MAX-DOAS instrument in Madrid certainly have the potential to provide interesting results on pollution in the city, and how it depends on emissions and meteorology. Such a study would then however be more appropriate for ACP than for AMT.

I also have some more detailed comments, which the authors could take into consideration when using the existing draft as base for another manuscript providing novel results and data.

Line 120: I am not sure that profile retrievals "try to reconstruct the photon paths" – in my view, they mainly try to find a vertical distribution that is consistent with the retrieved DSCDs

Table 1 / Table 2: I am not sure what exactly is meant by "All spectra and the Ring cross sections were allowed to shift and stretch (order 1) in wavelength". However, in my opinion, reference spectra should not be allowed to shift and stretch as they are measured at high precision. If the background spectrum (here: the zenith-sky measurement) is well calibrated using a Fraunhofer Atlas, the only spectrum that should be allowed to shift and stretch is the horizon measurement itself.

[Figure]

Line 239: Cloud clearance using AERONET data will work in the direction of the sun, but as far as I know, it does not guarantee 360° of cloud free measurements.

Figure 4 and discussion: I did not fully understand what was done here and why – surely, it does not make sense to use an atmosphere for the wrong surface height. I also fail to understand what the conclusions i) and ii) exactly imply, and how they follow from the fact that the profile retrieval is able to compensate a wrong atmospheric pressure profile by wrong extinction coefficients when reproducing O4 measurements.

Figure 9: I think it does not make sense to present two pieces of radial information from the onion peeling approach in this smoothed fashion that suggest a higher information content than there really is.

---

## Author Response (AR1)

We would like to sincerely thank the Reviewers for their support and constructive comments on the manuscript. Their comments have helped to improve the quality of our work. We provide here a detailed point-by-point answer (shown in blue), to their comments and suggestions.

**Reviewer 1:**

The present manuscript presents a complete analysis of O4 and NO2 vertical profiles during three months in Madrid, Spain with the aid of ground-based MAX-DOAS 2-D observations. The aerosol and NO2 vertical profiles in multiple viewing azimuth directions are presented here as well as the horizontal NO2 distribution around the measurement site. Finally, the 2-D MAX-DOAS NO2 near-surface concentrations are compared with the in-situ NO2 measurements in Madrid.

I recommend the publication of the manuscript after consideration of a major number of specific comments:

We thank the reviewer for her/his thorough and constructive comments, which we address below.

Specific comments:

1. Page 1, Line 19: Please write the spatial resolution of the mesoscale events.

We have included the spatial resolution (in the order of a few kilometers) in lines 25-26.

2. Page 1, Line 27: In my understanding, you used one inversion algorithm (not inversion algorithms) for the aerosol and the NO2. Please correct that and write the name of the inversion algorithm that is used (bePRO).

We have changed it by "an inversion algorithm" in line 19 in the abstract.

3. Page 1, Abstract: I would recommend that you write in a more clear way, the main findings of this study and the main contributions/innovations that you have made.

Thank you for this useful comment. We rewrote this part and we included in more detail the main findings of our study, from line 20 to line 24.

4. Page 2, Line 49: I would recommend to write that you have developed two MAX-DOAS instruments and not just MAX-DOAS instruments.

We developed one MAX-DOAS instrument, for this reason we specify now "we have deployed a Multi AXis Differential Optical Absorption Spectroscopy (MAXDOAS) instrument" in lines 63-64.

5. Introduction: It would be valuable to add a paragraph in which you cite previous MAX-DOAS studies of two-dimensional measurements (like Ortega, Schreier, Wang, Dimitropoulou etc.) as well as studies where MAX-DOAS observations are compared with in-situ measurements.

We have added a paragraph (lines 88-93) in which we cite previous studies that report measurement using MAXDOAS-2D instruments.

6. Section 3.2: Where do you expect to measure higher NO2 concentrations (North, South etc.)?

Based on previous studies, there is no clear, steady distribution of NO2 in Madrid. Instead there are strong spatial gradients and temporal changes (including considerable traffic hot-spots), thus making it difficult to predict with great accuracy how the NO2 will be distributed at a given time. However, mesoscale simulations in Madrid show that in general, higher NO2 mixing ratios are expected in the southern part of the city taking into account the population distribution and commuting patterns (see Picornell et al., 2019 for more details). We have included this issue in lines 214-222.

7. Page 7, Line 193: In your study, one complete MAX-DOAS scan takes one hour. The advantage is that you have a very nice horizontal sampling but at the other hand, you risk to measure the same NO2 air mass in multiple azimuthal directions (for example, during one hour, the NO2 that you observe in the North can be moved by the wind in the North East direction). Please add a sentence in which, you make clear the advantages and disadvantages of your choice.

Understood. We have added it in lines 264-271. Indeed, it will be useful for the reader to include the advantages and disadvantages of such measurements setup.

8. Page 11, Line 252: After the filtering of the MAX-DOAS measurements, which is the percentage of accepted scans?

We have included the percentage of cycles (slightly above 90 %) that were considered valid (concerning the quality checks) as input for the RTM. You can see this part in lines 349-352.

9. Page 11, Line 264: The RTM is the forward model and the bePRO is the inversion algorithm. Please correct this.

We have modified this part, (line 356 in our revised manuscript).

10. Page 12, Line 290: It's not exactly an analogous process because for the O4 and aerosol, non-linear calculations are performed and for trace gases as NO2, we have linear calculations. Please verify if it's the case for bePRO and correct or not this sentence.

Thank you for this appreciation. We have clarified that a linear analysis is made to estimate the vertical concentration profile of  $NO_2$  using the light paths derived from the non-linear analysis of the  $O_4$  and aerosol (from line 364 to line 372).

11. Page 13, line 310-318: You have used Standard atmosphere profiles, which are widely used in studies like the present one. But, you should include an uncertainty estimate of using a standard profile instead of a real profile (by meteorological measured data).

We have developed a more detailed uncertainty analysis. We have included the uncertainty sources in the whole analysis from line 482 to line 490. Concerning the use of a given atmospheric profile, we have concluded that the RMS of the relative variations (within the first 10 km height) was of about 8 %. We went a step further and estimated that, regarding the light paths, the RMS of the relative changes coming from the atmospheric profile choice was below 2 %.

12. Section 4.2: You should a paragraph in which you present an average error estimate of the retrievals and add a Table with all the error sources (smoothing error etc).

As described above, we have completed the section with the average uncertainties of the retrieval. A table has been included and appears in the text from line 493 to line 501.

13. Section 4.3: In your results, you should discuss the range of the estimated horizontal distances for the UV and Vis during your measurement period

The range of the estimated horizontal distances appear now in lines 517-519.

14. Figure 6: These results are from which measurement day and scan/hour? I assume that it is not the whole period, right?

Yes, these results are for the entire period, in line 540 it is marked that this correlation is for the entire campaign. We usually do this with the purpose of checking the goodness of the analysis for the entire campaign, it is a useful and rapid way to assess the simulations.

15. Figure 7: How do you explain the aerosol peak at around 50 deg. VAA and in high altitude?

This aerosol peak could come from traffic because there is a main road at this VAA. However, we are not sensitive above the boundary layer to know if this peak could be due to uncertainties in the RTM. Anyway, that would be one of the main ideas of this work: that the O4 DSCDs are the ones which drive the light path analysis. As shown in Section 4.2, an aerosol loading may cause a quite similar (or even the same) effect as small variations in the atmospheric profiles or parameters. However, this does not affect the light path estimation and the subsequent trace gas analysis, hence only affecting the certainty of assigning an irradiance extinction as aerosol (specially in higher layers), lines 565-569.

16. Page 20, Line 465: Why do you use the UV distance and the Vis which is larger?

We have mentioned in line X that we only take into account the air quality monitor stations which are at a distance from our MAXDOAS equal or lower than 10 km, and the UV light path ranges typically in the order of 8-10 km, hence that is why we chose the NO2 retrieved in the UV region for the comparison. It appears now in lines 658-662.

17. Figure 10: Please include a 1:1 line and put the same axis limits in both x, y axis in order to quantify rapidly the underestimation on the near-surface NO2 concentrations by the MAX-DOAS

Figure 11 (previously figure 10) has been modified in order to show 1 to 1 axis, so that the underestimation is easier to observe, as you suggest (line 672)

18. Page 21, Line 480: You write that the slope is lower than 1 (it is 0.4) which is true but you should add a sentence in which you discuss this finding. Is it in agreement with previous studies that compared MAX-DOAS and in-situ?

We have completed this part including some previous works in which similar conclusions were reached (we also discuss the slope value from line 686 to line 689).

19. Conclusions: You should make this section larger and discuss more your results

We have now a more complete summary and conclusions part (section 6).

20. Through the whole manuscript, references should be added, as I mentioned in previous comments

Several references have been added through the entire work.

Technical corrections

1. Page 2, line 34: gaseous pollutant concentrations instead of gaseous pollutants concentrations

Changed. Now line 44.

2. Page 3, line 73: path lengths instead of paths lengths

Changed. Now line 99.

3. Page 11, Line 256: inversion algorithm method instead of inversion algorithms.

Changed. Now line 356.

We would like to sincerely thank the reviewer for her/his support and comments on the manuscript. The comments have helped to improve the quality of our work and include some new information. We provide here a detailed point-by-point answer (shown in blue), to the comments and suggestions.

**Reviewer 2:**

In their manuscript "Two-dimensional monitoring of air pollution in Madrid, Spain using a MAXDOAS-2D instrument", the authors report on measurements in Madrid using a new MAX-DOAS instrument with both elevation and azimuth pointing capabilities. Examples of NO2 profile retrievals are discussed and some results of onion peelingretrievals presented. Finally, a comparison is performed between hourly mean values from the lowest MAX-DOAS profile level and data from the air quality network, showing good correlation. The manuscript is generally clear and well written but lacks detail in many places. It also does not provide reference to the many existing studies using similar instruments, performing similar retrievals, and addressing similar research questions.

My main problem with this manuscript is however the lack of novelty: In fact, I do not see anything new in this manuscript on instrument development, DOAS retrievals, profile retrievals, the onion peeling approach or the validation of the retrievals. The instrument is similar to many others operated (see Kreher et al., 2020), the DOAS retrieval is performed using the freely available software QDOAS, the profile retrieval is using the software BePro, the onion peeling follows the work by Ortega et al. And the validation is limited to a single figure showing measurements from a not further defined time period. I therefore unfortunately cannot recommend this manuscript for publication in Atmospheric Measurement Techniques.

The measurements of the 2d-MAX-DOAS instrument in Madrid certainly have the po-tential to provide interesting results on pollution in the city, and how it depends on emissions and meteorology. Such a study would then however be more appropriate for ACP than for AMT.

I also have some more detailed comments, which the authors could take into consideration when using the existing draft as base for another manuscript providing novel results and data.

We thank the reviewer for her/his comments, which we address below. We however think that AMT is the appropriate journal for publication of our results. To further add information on the capabilities of MAXDOAS-2D to the study of air pollution in Madrid, we have performed, and included in the revised manuscript, analysis of HONO spatial distributions. We now include an example of a twodimensional map of HONO at 6 UTC time for the same representative day we used for  $NO_2$ . To our knowledge, this is the first time in which a 2D instrument is used to retrieve the HONO spatial distribution. The DSCDs simulated and calculated are in good agreement, and the comparison has a slope of 1.12 and a correlation coefficient of 0.99. In addition, the MAXDOAS-2D measurement of HONO has added value for air pollution research in the city since it is not measured by the insitu monitors of the Council of Madrid air quality network. Therefore, our MAXDOAS-2D could provide some useful information regarding the mesoscale distribution of HONO, its role in the atmospheric chemistry in Madrid and its interactions with other trace gases such as NO2.

Line 120: I am not sure that profile retrievals "try to reconstruct the photon paths" – in my view, they mainly try to find a vertical distribution that is consistent with the retrieved DSCDs

Thank you. We have changed this description for the sake of clarity and we have added a better RTM summary (from line 160 to line 163).

Table 1 / Table 2: I am not sure what exactly is meant by "All spectra and the Ring cross sections were allowed to shift and stretch (order 1) in wavelength". However, in my opinion, reference spectra should not be allowed to shift and stretch as they are measured at high precision. If the background spectrum (here: the zenith-sky measurement) is well calibrated using a Fraunhofer Atlas, the only spectrum that should be allowed to shift and stretch is the horizon measurement itself.

Thank you for this comment. We think we failed to provide a clear explanation in our original submission. We only let to shift the measured spectra (with the MAXDOAS-2D) and the Ring, not the spectral absorption cross sections of the trace gases. We decided to include a shift to the Ring cross section because it is based on the inelastic rotational Raman scattering, which slightly changes the wavelength of the scattered photon when the scattering occurs, so it should have a little shift to improve the analysis. We checked the values of the Ring shift and although low, it improved the analysis, so we think that we could let the Ring shift in wavelength. This is now clarified in Tables 1 and 2.

Line 239: Cloud clearance using AERONET data will work in the direction of the sun, but as far as I know, it does not guarantee 360° of cloud free measurements.

We have added more information regarding the role of cloud measurements in our study. We mention the AERONET data because we compared the AERONET data with our MATLAB code data and the results are similar. Now, we have added the MATLAB code filter that we programmed from scratch (it is explained from line 312 to line 332).

Figure 4 and discussion: I did not fully understand what was done here and why -surely, it does not make sense to use an atmosphere for the wrong surface height. I also fail to understand what the conclusions i) and ii) exactly imply, and how they follow from the fact that the profile retrieval is able to compensate a wrong atmospheric pressure profile by wrong extinction coefficients when reproducing O4 measurements.

We would like to take the opportunity to clarify that we did not use a wrong surface height, in which case we agree it would not make sense. We have used a height grid of layers that start right at the surface (0 m height). What we did was to interpolate the US Standard pressure profile (that is assumed to be accurate for the sea level) to the mean height of Madrid above sea level. Using those two sets of atmospheric profiles as examples, we ended up having very similar simulated DSCDs of  $O_4$  in both cases, hence it seems that small variations in the atmospheric profiles do not affect significantly the  $O_4$  analysis, thus we concluded in i) that the main driver of the  $O_4$ retrieval are the measured  $O_4$  DSCDs, which gives confidence to the overall analysis. However, each set of atmospheric profiles gave rise to notable differences in the extinction coefficients (especially above the surface layer). Therefore, we concluded that variations in physical parameters such as the pressure profile can produce changes in the extinction coefficients, hence given the difficulty to obtain very accurate atmospheric profiles, we think that as of now we cannot reliable assign those extinction values as particulate matter extinction (i.e. to aerosols). We prefer to discuss uncertainties in the atmospheric profiles rather than true or false profiles. Nonetheless, as shown in Figure 5, the fact that the simulated DSCDs still reproduce with high accuracy the measured O4 DSCDs means that the light paths derived will be essentially the same (regardless the chosen atmospheric profile), and hence will ultimately generate almost the same results for the trace gases profiles.

Figure 9: I think it does not make sense to present two pieces of radial information from the onion peeling approach in this smoothed fashion that suggest a higher information content than there really is. We tried to specify within the text that we carried out the calculations with two radial values, we decided to show the contour because we thought it would be easier to grasp both NO2 location and its temporal variation at a glance. However, we understand the reviewer's point that interpolating from just two radial values may be misleading. Hence we have modified the figure in our revised manuscript to present our results through an usual polar plot without interpolation (see lines 638-641 for the figure caption).

```
         Two-dimensional monitoring of air pollution in Madrid using a
                               MAXDOAS-2D instrument
    David Garcia-Nieto1, 2, Nuria Benavent1, 2, Rafael Borge2 and Alfonso
    Saiz-Lopez1
    1 Department of Atmospheric Chemistry and Climate, Institute of
    Physical Chemistry Rocasolano, CSIC, Madrid 28006, Spain
    2 Universidad Politécnica de Madrid, UPM, 28006 Madrid, Spain
    *Corresponding author: Alfonso Saiz-Lopez (a.saiz@csic.es)
    Abstract
          Trace gases play a key role in the chemistry of urban atmospheres.
    Therefore, knowledge about their spatial distribution is needed to
    fully characterize the air quality in urban areas. Using a new Multi-
    AXis Differential Optical Absorption Spectroscopy (MAXDOAS)-2D
     instrument, along with an inversion algorithm (bePRO), we report the
```

first two-dimensional maps of nitrogen dioxide (NO2) and nitrous acid 20 (HONO) concentrations in the city of Madrid, Spain. Measurements were 21 22 made during two months (May 6 -July 5 2019) and peak mixing ratios of 12 ppbv and 0.7 ppbv for NO2 and HONO, respectively, were observed in 23 the early morning in the south-pointing geometry. We found good general 24 agreement between the MAXDOAS-2D mesoscale observations -which provide 25 a typical spatial range of a few kilometers- and the in-situ 26 measurements provided by Madrid's air quality monitoring stations. In 27 addition to vertical profiles, we studied the horizontal gradients of 28 29 NO2 in the surface layer by applying the different horizontal light path lengths in the two spectral regions included in the NO2 spectral 30

analysis: ultraviolet (UV, at 360 nm) and visible (VIS, 477 nm). We also investigate the sensitivity of the instrument to infer verticallydistributed information on aerosol extinction coefficients and discuss possible future ways to improve the retrievals. The retrieval of twodimensional distributions of trace gas concentrations reported here provides valuable spatial information for the study of air quality in the city of Madrid.

1 Introduction

Air pollution in urban areas has become a concern in our society 41 because it represents a major risk to human health and the environment 42 (WHO, 2019). Air quality is often expressed as the state of air 43 pollution in terms of gaseous pollutant concentrations as well as size 44 and number of particulate matter that may affect human health, 45 ecosystems and climate (Monks et al., 2009). Integral understanding 46 47 of air pollution requires knowledge about the sources, pollutants, chemical composition and spatial distribution, and their transport 48 phenomena in the atmosphere (EEA, 2019). 49

Madrid, Spain, has suffered from severe air pollution in recent years, with episodes of large nitrogen dioxide  $(NO_2)$  and ozone  $(O_3)$ 52 concentrations. In an effort to control and reduce high pollution 53 events, the local government has enforced some traffic restriction 54 measures (Izquierdo et al., 2020) and has set up several in-situ air 55 quality monitoring stations over the city's metropolitan area. These 56 in-situ instruments -as of today- cannot measure some important trace 57 present in the atmosphere and their values are only 58 gases representative of the immediate surrounding of the instruments and at 59 surface level. There is therefore a need for mesoscale analysis (both 60 in horizontal and vertical) of urban air pollution that could 61 62 complement the in-situ measurements. With this aim, we have deployed a Multi AXis Differential Optical Absorption Spectroscopy (MAXDOAS) 63 instrument for air pollution measurements in Madrid. MAXDOAS is a 64 widely used technique for the detection of trace gases in the 65 atmosphere and it is based on the wavelength dependent absorption of 66 scattered sunlight by atmospheric constituents (Platt and Stutz, 67 2008). In addition to routinely monitored, regulated species such as 68  $NO_2$  and  $O_3$ , MAXDOAS provides mesoscale measurements of other trace 69 gases that are relevant to understand atmospheric chemistry, such as 70 nitrous acid (HONO), formaldehyde (HCHO) or glyoxal (CHOCHO). Over the 71 past few years, we have reported trace gas measurements in Madrid 72 using the MAXDOAS technique (Wang et al., 2016; Garcia-Nieto et al., 73 74 2018; Benavent et al., 2019) as well as pollutants trend analysis and chemical transport modelling (Borge et al., 2018; Cuevas et al., 2014; 75 Saiz-Lopez et al., 2017). 76

For this work, a new two-dimensional MAXDOAS instrument (which 78 will be described in Sect. 3 and will be hereafter referred to as 79 MAXDOAS-2D) has been built, tested and set up to take continuous 80 measurements in Madrid. This instrument represents a follow-up 81 82 development to our previous one-dimensional instrument (MAXDOAS-1D, see Wang et al., 2016) that incorporates the capability of moving in 83 84 the azimuthal dimension, therefore allowing the collection of spectra pointing at any angular direction. This additional capability allows 85 the measurement of both the horizontal and vertical trace gas (e.g. 86  $NO_2$ ) distribution throughout the city and in turn the generation of 87 two-dimensional maps of trace gas concentrations. Several works using 88 two-dimensional MAXDOAS instruments have been carried out in recent 89 years (e.g. Ortega et al., 2015, Yang et al., 2019, Schreier et al., 90 2019, Dimitropolou et al., 2020). These studies were mostly focused 91 on mapping the NO2 distribution in urban environments and assessing 92 93 its role for air quality monitoring.

Here we present two months of MAXDOAS-2D measurements of scattered sunlight spectra. The measurements were taken from May 6, 96 2019 to July 5, 2019, with focus on the evaluation of  $NO_2$  vertical 97 concentration profiles and the characterization of horizontal light 98 path lengths. We will also provide the retrieval of HONO as an example 99 of the potential of the MAXDOAS-2D measurements. This represents the 100 101 first two-dimensional MAXDOAS measurements in Madrid. An assessment of the relation between the MAXDOAS analysis and the in-situ 102 instruments in the city was carried out. Sect. 2 provides details of 103 the DOAS technique while Sect. 3 describes the experimental setup. The 104 inversion methods and the atmospheric parameters chosen for the 105 106 analysis is detailed in Sect. 4. The two-dimensional  $NO_2$  and HONOdistributions, an evaluation of the light path geometries, along with 107 their relative probabilities, and an assessment of horizontal mixing 108 109 ratio gradients near the surface are discussed in Sect. 5. Finally, Sect. 6 contains conclusions and possible future work. 110

**112 2 Brief introduction to the DOAS method**

The absorption spectroscopy field has been developed for several decades within different research disciplines (such as remote sensing, astronomy or atomic and molecular physics). Its foundation relies on the absorption of radiation when interacting with a certain sample. The 
[revised manuscript text omitted]

---

## Author Response (AR3)

We would like to sincerely thank the Reviewers for their support and constructive comments on the manuscript. Their comments have helped to improve the quality of our work. We provide here a detailed point-by-point answer (shown in blue), to their comments and suggestions.

**REFEREE 1**

Concerning the revisions, I suggest a more detailed discussion in Section 5.3, which can be the core of the results and bring new ideas in this study.

Based on this suggestion and others from Referee 3, we have added some more details to this specific section (from line 499 to line 521).

Technical: (Page 20, line 495) --> The

Corrected (now in line 400).

**REFEREE 2**

The authors have reacted on most of the detailed suggestions made by myself and another reviewer. They also have added a short section on one scan of HONO retrievals.

Unfortunately, my main point remains unchanged: I do not see what is new in this manuscript with respect to measurement techniques or retrievals, the topics covered in AMT:

* the instrument's ability to measure in several azimuths has been state of the art for MAX-DOAS instruments for many years now

\* the DOAS retrieval code used is well documented in previous publications

\* the profile retrieval algorithm used is well documented in previous publications

\* the onion peeling approach using UV and visible NO2 retrievals was already published and used in several publications

The value of these measurements is in providing information on pollutants in Madrid, and here I do see potential. However, AMT is not the right journal for the presentation of measurement results, and much more data and analysis are needed in order to make this a valuable manuscript for ACP or a similar journal.

We thank the reviewer for the time devoted to read through the paper. We think the paper falls within the scope of AMT: "*The main subject areas comprise the development, intercomparison, and validation of measurement instruments and techniques of data processing and information retrieval for gases, aerosols, and clouds. Papers submitted to AMT must contain atmospheric measurements, laboratory measurements relevant for atmospheric science, and/or theoretical calculations of measurements simulations with detailed error analysis including instrument simulations*".

Our paper contains atmospheric measurements that report the spatial (horizontal and vertical) distribution of $NO_2$ and HONO, not typically measured by air quality networks, in a European capital, particularly affected by air pollution. Although, the DOAS technique and the corresponding analysis theory is already well published in the literature, we think that its application, comparison with in-situ instruments, and its contextualization about the potential of mesoscale spatially-resolved measurements in air quality research in Madrid further adds to the literature on the potential of MAXDOAS instruments for air pollution research.

On a more technical side, we think that Section 4.2 and the analysis on the sensitivity of the aerosol profile retrieval to different atmospheric profiles can be of interest in the assignment of MAXDOAS derived extinction coefficients to aerosols, particular for layers above the boundary layer.

This paper presents nitrogen dioxide and nitrous acid observations over Madrid obtained using a new 2D MAXDOAS instrument. The reported observations show good agreement with mesoscale observations and in-situ measurements provided by Madrid air quality monitoring stations. The capacity of this 2D MAXDOAS instrument to infer horizontal gradients and vertical distribution is discussed as well as future strategies to improve retrievals. The paper is clearly written and the MAXDOAS-2D datasets presented here are a valuable addition to understand air quality in Madrid. For those reasons the paper is worth publication.

However, before publication this paper would benefit of more specific descriptions of technical aspects of RTM calculations and inversion methods. Large extensions of the text are devoted to general descriptions of the theory and methods employed by the authors but if provides insufficient details about the specifics of RTM calculations, inversion algorithms, and protocols used to compare in-situ and MAXDOAS-2D observations. Someone trying to reproduce the results presented here will have a very hard time given the lack of specific details. The minor comments section below specifies some of the sections where further details will be useful.

Thank you for the constructive comments. We have now remove some sections of the text devoted to general descriptions of the theory and methods, and added more details the specifics of the data analysis and results.

Minor comments:

Cloud screening algorithm: How is AERONET information combined with the photos taken by the camera. AERONET observations follow the Sun so how they do contribute to other azimuth angles. It may help to understand better this step and increase confidence in its efficacy to include a plot or a discussion of the cloud screening statistics.

We have added more information regarding the cloud filtering that combines the AERONET observations and the pictures taken by the camera installed in the MAXDOAS-2D. The discussion of the cloud screening statistics is also explained in more detail (from line 269 to 274).

Section 4.2 provides a nice description of the general aspects of profile inversion in MAX-DOAS retrievals but is probably not necessary in a research paper. Particularly if we consider that details about the forward model and the inversion algorithms are limited to one single sentence referencing Clémer et al., 2010. Further technical details would be more helpful to the specialized reader than the general description provided.

We have now removed the general aspects of the profile inversion and we have included more information about the RTM parameters that we have used (from line 286 to line 430).

Since the retrieval is using US Standard atmosphere as unique source of p/T profiles it will be interesting to know how different it is from the typical p/T profiles in Madrid during the observation campaign.

We have discussed about it in lines 389-396. There, we explain that we have carried out several tests concerning the atmospheric profiles. For instance, we took the average surface temperature and pressure values during the campaign (May-July, 2019) and included them into the retrievals. We then constructed the air density vertical profile and evaluated the relative variations with respect to the same air density profile in the U.S. Standard Atmosphere, and found that (within the first 10 km height) the RMS of such relative variations was of about 8 %. It was a small but not negligible change, therefore we decided to push the tests forward, assessing the relative changes produced in their respective light paths. In that regard, we found that the RMS of the relative changes under the same height (i.e. 10 km) were below 2 %, thus we concluded that the main driver of the light path retrievals is the measured $O_4$ DSCDs and that we could safely chose the U.S. Standard Atmosphere for our studies.

The RTM calculations in the estimation of NO2 horizontal gradients are performed for each VEA or only for VEA = 0. If so, the onion peeling approach is only applied to surface layer mixing ratios?

We have completed this part, from line 502 to line 504, indicating that we are using the measured $NO_2$ DSCDs at a VEA 1 degree.

Section 5.4 compares in-situ and MAXDOAS-2D NO2 observations looking at the correlation between surface layer retrievals and in-situ observations. While the results here show good correlation they bring little confirmation about the 2D capabilities of the instrument (which is the novel capacity that is presented in the paper) because the comparison is done averaging results over an hour. Would it be possible to analyze qualitative agreement between in-situ and MAXDOAS-2D as function of VAA? How many in-situ stations are finally considered in the comparisons out of the 24 available? Do correlations shown in figure 10 increase or decrease when MAXDOAS VAA is considered in the analysis.

Thank you for this useful comment, we have included in Section 5.4 the remaining stations (line 537). If we divide the azimuthal lap in slices of 20º width for some of them we end up having just one monitoring station, hence we considered that to not be statistically solid, for this reason we done the comparison with the $NO_2$ surface layer hourly-averaged data.

Technical comments:

Line 24: south-pointing geometry is meaningless without providing an origin. Is this looking towards Madrid's outskirts or towards downtown?

We have marked that is in the southern part of the downtown area (lines 21-22).

Line 61: add dimensions after "both in horizontal and vertical "

Added (line 51).

Line 68: remove comma after monitored

Removed (line 58).

Line 99: remove "will"

Removed (line 83).

Table 1 and table 2: Most likely due to my lack of knowledge but I wonder if Serdyuchenko et al., at 223K was fitted twice or it is a typo.

Thank you, we have an error in the tables 1 and 2. So we have corrected the ozone cross section temperature, one is 273 K and the second one is 223 K.

Figure 3: Would it be possible to include the viewing geometries corresponding to the shown DSCDs fits?

Thank you, the viewing geometries are now included (Figure 3).

Figure 4: It seems that the results presented here are for the whole campaign. Please clarify? How is surface extinction coefficient defined? It would be helpful to illustrate in some way the vertical grid of the retrievals.

We have clarified that this comparison was carried out for a clear sky day (Figure 4.). And, we have included the definition of extinction coefficient (line 364).

Figure 5: I guess is also for the whole period of time.

We have added this clarification (Figure 5).

Line 495: Missing "T" at the end of the line.

Done. Thank you (now in line 400).

Line 536. Vertical profiles are retrieve not only using the RTM explained in section 4. That is one of the components of the inversion algorithm. Besides, details of the RTM are not provided in section 4.

The RTM used in Section 5.1 is bePRO, which has been widely described in previous publications. Nevertheless, we provide a brief description and point to the appropriate references in Section 4.

Figure 6 caption can be expanded to provide further details. What do the red dots represent? It must be some kind of average over the campaign. Besides that, red dots are not simulated or otherwise the x-axes labels "DSCD measured" don't make much sense.

We have included more details regarding the meaning of the graphs (Figure 6).

Figure 10: What represents the radial dimension of the polar plot. The estimated distance from the measurement center? If so, is it the mean distance of each layer with respect to the center?

We have completed the caption of the Figure including the meaning of the symbols (Figure 10).